# Impact of COVID-19 on Organizational Support in Financial Technology

Christian Herdinata [1,*] and Fransisca D. Pranatasari [2]

1 School of Business and Management, Universitas Ciputra Surabaya, CitraLand CBD Boulevard, Surabaya 60219, Indonesia
2 Faculty of Economics, Sanata Dharma University, Yogyakarta 55281, Indonesia; fr.desiana@gmail.com
* Correspondence: christian.herdinata@ciputra.ac.id

**Abstract:** The purpose of this research is to determine the organizational support and role played by MSME owners who have businesses based on financial technology during the COVID-19 and earlier periods. It also aims to determine the priorities that need to be considered in organizations in various conditions through fintech-based training. This is an experimental research with training given to the experimental group and none to the control. A paired-samples *t*-test analysis technique was used to determine the significant differences in the same group under different conditions. The result showed that fintech-based training is important in supporting organizations in the business groups that received training and those that have significant differences. Therefore, this training is needed, especially during crises such as the COVID-19 pandemic.

**Keywords:** financial technology; organizational support; experimental design; COVID-19 pandemic

## 1. Introduction

This research highlights the role of organizational support in helping fintech-based MSMEs survive in uncertain situations such as the COVID-19 pandemic. Financial technology has become increasingly popular since the 1990s, with strong global development in the financial industry (Khuong et al. 2022). The worldwide acceleration of technology growth has also impacted fintech-based business strategies. Several countries in Southeast Asia, including Indonesia, are potential markets for fintech growth (Tut 2020). The rapid changes in the digital world and the COVID-19 pandemic require MSMEs to improve and strengthen dynamic capability activities to achieve a competitive advantage in innovation performance among organizations (Cheng et al. 2014). Southeast Asia is the second-most affected region by the pandemic, with 26% of positive cases and a 19% death rate as of August 2020 (WHO 2020). The sudden inception of the virus did not only impact the global death rate but adversely affected the economic shock of MSMEs (Juergensen et al. 2020). This economic growth imbalance has been experienced globally during previous pandemic conditions, such as the early discovery of HIV, AIDS, and other diseases (Goodell 2020; GPMB 2019; Santaeulalia-Llopis 2008; Yach et al. 2006). Due to this pandemic, the USA has lost over 500 billion US dollars, or 0.6% of its global income (Fan et al. 2018).

According to Goodell (2020), the pandemic is leading to a long-term change in shopping behavior worldwide. This is similar to the decrease in domestic spending and demand during the HIV/AIDS epidemic (Goodell 2020; Santaeulalia-Llopis 2008). The COVID-19 pandemic has made the economic environment even more challenging, with an emphasis on sustainability. This is a big challenge for the global economy during disasters (Bjørnskov 2008; Cavallo et al. 2013; Fukuyama 1995; Ghesquiere and Mahul 2010; Noy 2009). Furthermore, the pandemic has resulted in significant geographical shifts in supply and demand, which in turn has created problems for finely tuned global supply chains. In the extreme, this supply chain disruption is felt in almost all countries in the world such as the



US, Europe, China, and even emerging markets (www.weforum.org, 2022 last accessed on 16 February 2022).

Among the main problems is the unprecedented pressure on global supply chains and the series of lockdowns and restrictions that vary in time and severity from one country to another. The pressure on supply chains in emerging markets, dominated by MSMEs that support the country's economy, which has an impact on domestic economic stability. For this reason, business organizations have to be resilient and capable of adapting to major disruptions in order to develop long-term strategies and solutions to complex challenges. Agile organizations that use technology will be able to withstand challenges associated with the COVID-19 pandemic. This is because technology has created new opportunities for digital financial services to accelerate and increase financial inclusion, amid social distancing and containment measures. At the same time, the risks that emerged before the pandemic, as digital financial services evolved, are becoming increasingly relevant (Mikalef and Pateli 2017; Sahay et al. 2020; Tut 2020).

Among the positive effects for financial institutions and the adoption of fintech by consumers is the growth in the use of digital payment media (Sahay et al. 2020; Tut 2020). Fintech in Indonesia still has a lot of development opportunities. This can be illustrated by the emergence of the Indonesian Fintech Association (AFI) in 2015, which attracted business owners to provide trusted and reliable partners to build the fintech ecosystem. Currently, 30% of companies in Indonesia are already using fintech, which has shown significant growth from 7% in 2006 and 2007 to 78% in 2017 with 135 to 140 companies. FinTech Profile in Indonesia by Sector in 2017 is Personal Financial planning 8.15%, crowdfunding 8.15%, lending 17.78%, aggregator 12.59%, payment 42.22%, and others 11.11% (http://www.ibs.ac.id, 2017 last accessed on 16 February 2022). With this rapid development, it is estimated that the development of fintech in Indonesia will continuously increase. Organizational agility has been shown to help build competitive advantage in an uncertain environment empirically related to the dynamic capability supporting IT and competitive performance (Mikalef and Pateli 2017). Organizations are faced with significant uncertainties not limited to national, economic, or social boundaries. This crisis is the first test of resilience to fintech-based businesses that are quite risky in building the digital infrastructure of resources during the pandemic (Carnevale and Hatak 2020; Sahay et al. 2020). The fintech landscape is still considered competitive in the post-COVID-19 era. Therefore, organizations need appropriate strategic planning to manage changes in their business cycle (Côrte-Real et al. 2019).

For this reason, it is necessary to have organizational support for fintech businesses by reducing business operating costs, improving customer service, enhancing distribution channels, reaping operational benefits, providing an effective support role for operations, and increasing competitive ability (Grandon and Pearson 2004; Akgunduz et al. 2018). Therefore, fintech business SMEs need to be equipped with fintech-based training to deal with the pandemic. The assumption is that training is designed to achieve strategic maneuvers to enhance organizational performance (Van Eemeren and Houtlosser 1999, 2006). This strategic maneuver is proven to improve business performance (Fahed-Sreih and El-Kassar 2017). In normal conditions, organizational support in fintech efforts increases performance and overall productivity (Akgunduz et al. 2018; Caesens and Stinglhamber 2014; Jeung et al. 2017). The perceived effect of organizational support on fintech businesses can increase the proactive work personality of both employees and MSME owners (Akgunduz et al. 2018). The pandemic, directly and indirectly, impacted economic aspects, which has consequences for future financial investigations (Goodell 2020).

Research on the pandemic discusses more medical problems that are at the forefront. On the other hand, its impact is broader, with the emergence of higher inequality as a longer-term legacy. Limited studies have provided an in-depth solution to the economic crisis due to the pandemic (Bloom et al. 2018; Goodell 2020). Almost all regions of the world have not fully recovered from their economic situation. Germany, Japan, Turkey, UK, USA, Brazil, India, Indonesia, Eurozone, China and many more, had a similar condition with

drastic decline in Economic Growth in Q2 2020 (WHO 2022). Concerns about the economic growth of business organizations are estimated to be a global revenue risk (Fan et al. 2018). Given the impact of the virus, business organizations should not underestimate this and need to make the right decision (Bloom et al. 2018; Fan et al. 2018; Goodell 2020; GPMB 2019; Santaeulalia-Llopis 2008; Tam et al. 2016; Yach et al. 2006). India, whose economy doubled during the pandemic, has a strong positive trend, with a forecast of 8.5% growth in the coming year. It is estimated that Asia will experience global growth in 2022 at a significant rate. Therefore, Indonesia also needs to prepare its MSMEs to keep up with the accelerated growth rate to improve its economic conditions.

The core concept of organizational support in fintech amidst the pandemic is also new business organizations that underwent dramatic changes and adaptations. Similar conditions occurred during the emergence of Ebola in West Africa from 2013 to 2016 (Fan et al. 2018; Tam et al. 2016) and HIV AIDS (Goodell 2020; Santaeulalia-Llopis 2008). Fortunately, the digitalization era is creating significant turbulence in all aspects of the business, including financial technology (Grandon and Pearson 2004; Leng et al. 2018). Lack of social trust leads to additional transaction costs throughout the financial system (Bjørnskov 2008; Fukuyama 1995). Inconsistency in the findings related to solutions for handling organizational and economic aspects is an opportunity to carry out in-depth analysis, especially with MSMEs. It is hoped that with fintech-based training, MSME businesses can survive during the pandemic. According to Ismail and Zain (2015), not all trainings significantly impact their students. This insignificant training result is probably due to the ineffective training design for the target trainees (Ismail and Zain 2015). Several facts have stated that the pandemic affects different portions of the economic spectrum, and therefore, it is important to design the right solution (Noy 2009).

Countries should immediately arrange a solution to the conditions of the COVID-19 pandemic to regain their people's social trust because it can potentially become a disaster (Ghesquiere and Mahul 2010). Solutions are designed to achieve a competitive advantage in strategic performance, including the ability to have market share and success compared to competitors (Côrte-Real et al. 2019). This research was conducted to prove that the fintech-based training provided can be a solution for MSMEs to survive the pandemic. Organizations supported by the right mechanism enable them to survive in any condition. The solution is to design fintech-based training related to financial literacy, fintech regulation, collaboration in the ecosystem, and analyze the different effects of training given to MSME owners. This study focuses on MSME owners who have businesses based on financial technology by investigating the role of organizational support during and before the pandemic. The objectives of this study are as follows:

1. To investigate the effects of fintech-based training given to MSME owners.
2. To determine the differences in the role of organizational support in fintech efforts in various conditions before and during the pandemic.

This research will benefit owners of fintech-based MSMEs where organizational support is important in supporting their survival capacity. Another benefit is to provide recommendations for the government to ensure the financial impact of the pandemic. This is because the economic benefits of fintech can accelerate the transmission of monetary policy, help increase the speed of money circulation, and overall economic growth. Based on the results, this research provides recommendations for stakeholders to develop a fintech-based training mode related to organizational support in dealing with the pandemic. This includes aspects of regulation, collaboration, and literacy as additional skills for MSME owners to survive. The training could be focused on the priority factors of capitalizing market reconfiguration (MCR) and operational adjustment reconfiguration (OAR).

## 2. Literature Review and Hypothesis

Limited studies have been conducted on the possible link between pandemic issues and financial technology-based businesses. The theory of planned behavior (TPB) is the basis of research that can predict and explain the behavior of various domains, including

the use of technology (Grandon and Pearson 2004). TPB was developed to explain behavior in different domains in various contexts. It supports a causal relationship between perceived strategic value and individual behavioral adoption. In general, perception affects intention, and individual behavior, which is this intention becomes the central factor in TPB and ultimately decides to adopt a certain behavior (Ajzen 1991; Grandon and Pearson 2004). Social changes in human behavior explain that an organization must be designed to meet human expectations to positively impact the organization (Akgunduz et al. 2018). Employees exhibit positive behavior dependent on behavioral control, for example, the availability of perceived opportunities and resources to perform the behavior and contribute to their organization (Steinmetz et al. 2011; Ibrahim et al. 2016). Organizational support is believed to generate effective commitment through achieving various things, especially in the development of fintech for agility and a greater survival rate. Based on the norm of reciprocity, the organizational treatment creates a sense of obligation that motivates employees to help achieve their goals. In relation to this, organizational support is considered a motivational construct that helps employees become physically engaged in tasks and empathically connect people with their job requirements (Caesens and Stinglhamber 2014; Ibrahim et al. 2016; Jeung et al. 2017). The training program was the most powerful activity among many organizational interventions (Guzzo et al. 1985).

Over the past two decades, the world has faced several contagious disease outbreaks, such as Ebola, Influenza A (H1N1), SARS, MERS, and the Zika virus, with the most recent being COVID-19. These outbreaks have had an enormous global impact on economic disruption on local and global public resources and human health. Economic prospects and the quantity and quality of employment are deteriorating rapidly (Juergensen et al. 2020). The treatment of the pandemic conditions is dissimilar to the event of a global nuclear war which impacts national defense and is relevant only in the cost sector. This is because its effect was much more slowly but created an overflow and market reaction leading to a direct, damaging global economic impact in every region of the world (Goodell 2020; Inegbedion 2021). It is a very large disaster that has a negative effect on output both in the short and long term. Developing countries usually suffer more when experiencing natural disasters, and therefore they need to prepare preventive economic activities that support MSMEs (Noy 2009; Cavallo et al. 2013).

Organizations must remain vigilant and adapt to unexpected events, such as external crises, which create increased uncertainty among their workforce and pose an immediate threat to performance and viability (Carnevale and Hatak 2020). According to preliminary studies, organizations are required to be able to carry out a company's potential strategic response to the crisis, through savings, persistence, innovation, and exit. There are many integrated ways to address the COVID-19 challenge for an organization (Buheji and Buheji 2020; Carnevale and Hatak 2020). Digitalization has had a number of significant impacts on the management design of a business organization. Its development has covered various aspects, among which is finance, which is changing the landscape of the sector, profoundly increasing access to its services (Sahay et al. 2020). The increasing number of MSMEs in developing countries is based on financial technology because they are considered to be a strong economic driver with high flexibility capabilities in their various business conditions (Ahmedova 2015; Kartiwi and MacGregor 2011; Mittal et al. 2018; Zhou 2016). MSMEs are also required to be able to identify new external and internal empowerment of business creation in various conditions.

At the beginning of the pandemic, there were indications of a negative impact on fintech adoption, but favorable short-term regulatory changes have reversed some of these effects. The use of all electronic payment cards decreased significantly during the pandemic except for credit cards because consumers switched to cheaper forms of payment. Furthermore, due to the increase in risk of transmission, many payments have turned to electronic. The flow through the fintech platform significantly affects the global contraction of economic activity (Tut 2020). These RegTech capabilities can effectively provide specific insights into the spread and prevention of fraud. Fintech provides economic benefits,

which accelerates the transmission of monetary policy and help increase the speed of money circulation (Turki et al. 2020).

In a competitive business environment, organizations are required to be able to increase creativity. The era of financial technology demands a more competitive organization and creativity to find solutions to all their problems. Organizational support design also aims to increase job satisfaction due to behavior that stimulates mutual emotions (Akgunduz et al. 2018; Caesens and Stinglhamber 2014; Ibrahim et al. 2016; Jeung et al. 2017; Leng et al. 2018). When organizations value their employees more, organizational behavior will also be more positive. Based on this theory, developing and maintaining relationships between individuals is based on utility and finance (Çakar and Yildiz 2009; Park et al. 2016). Other factors are respecting employees 'contributions to their welfare, identifying extra efforts, responding to complaints, paying attention to their job satisfaction and pride (Akgunduz et al. 2018). Organizational support is needed in carrying out as follows: (1) reducing costs of business operations, (2) improving customer service, (3) enhancing distribution channels, (4) reaping operational benefits, (5) providing effective support role to operations and (6) increasing ability to compete.

The COVID-19 crisis has had numerous impacts on the economic sector with difficulty in predicting its large scale impact (Goodell 2020). Organizations suddenly have to face something unprecedented and are required to find new solutions to the challenges that arise in many areas of their operations (Carnevale and Hatak 2020). All parties, including UMKM owners, are forced to face an uncertain phenomenon. This is in addition to MSME owners, who need to find solutions to avoid financial loss. Therefore, in order to survive the pandemic, MSMEs must maintain or improve their business performance to achieve organizational goals. Hafidhah and Martono (2019) stated that good performance can be achieved with organizational support. Special training is needed to support MSMEs because not all understand how to design a good organizational support. The hypotheses formulated for the study are as follows:

**Hypothesis 1 (H1).** *There are differences in organizational support in fintech efforts in the conditions before and during the pandemic.*

**Hypothesis 2 (H2).** *There are differences in organizational support with and without fintech efforts before and during the pandemic.*

**Hypothesis 3 (H3).** *There are differences in organizational support in fintech efforts and training before the pandemic.*

**Hypothesis 4 (H4).** *There are differences in organizational support in fintech efforts in the conditions during and before the pandemic.*

**Hypothesis 5 (H5).** *There are differences in organizational support for fintech efforts in the conditions during the pandemic and without training and before the pandemic.*

A summary of the literature presented in this section has discussed the urgency of MSMEs as business organizations to undergo extreme changes. The core concept of organizational support in fintech efforts amidst the pandemic is relatively new. However, none of these studies were aware of the analysis using an experimental approach through training. This is in line with the theory of planned behavior, where, through training, a person will exhibit a certain behavior. As a replacement, organization support reduces business operating costs, improves customer service, enhances distribution channels, obtains operational benefits, provides an effective support role for operations, and improves competitive ability. Furthermore, fintech collaboration and literacy were tested before and during the pandemic. Therefore, there are gaps in the literature that need to be filled. The following sections will present the data and methods used in this investigation.

## 3. Methodology

According to preliminary studies, the perceived strategic value of e-commerce tends to have a significant impact on managers' attitudes toward e-payment adoption, with organizational support and managerial productivity as the most influential variables (Grandon and Pearson 2004). Fintech-based organizations need a strong organizational support role to survive the pandemic. Therefore, this research recommended organizational support as the main consideration for Indonesian MSMEs in carrying out their business. Participants were business owners in developing countries who were determined using the purposive sampling technique (Bank Indonesia 2015).

The questionnaire method used for data collection was adopted from Grandon and Pearson's (2004) research regarding organizational support, with a five-point Likert scale from strongly disagree to agree used for analysis. This study used a quasi-experimental design (Chang and Chen 2015) to explore the effect of organizational support on MSMEs' resilience in facing various conditions before and during the pandemic and the fintech-based training.

### 3.1. Experimental Design

Experimental design is a careful balance of several features, including "strength," "generalizability," various forms of "validity," "practicality," and "cost". In an experimental design, the following criteria are necessary at least to ensure that it is reliable in interpreting the data: (1) the design point should have an equal influence on the determination regression coefficient and effect estimate; (2) it must be able to detect the need for nonlinear terms; (3) it must be strong against model specification errors; and (4) it should be built to provide appropriate information for a follow-up test. Ryan and Morgan (2007) stated that the experimental technique is necessary to ensure process control techniques are routinely applied to the control variable, regardless of whether the experiment is being conducted.

The research sample is 100 fintech-based SMEs (based on SME Criteria on Table 1) analyzed in the following control and experimental groups:

a.     Group A consisted of 100 MSME players before the COVID-19 pandemic.
b.     Group B comprised 100 MSME players during the pandemic.
c.     Group C consisted of 46 MSME actors who received fintech-based training on organizational support during the pandemic.
d.     Group D consisted of 46 MSME actors who did not receive fintech-based training on organizational support during the pandemic.
e.     Group E comprised 46 MSME actors who received fintech-based training on organizational support before the pandemic.
f.     Group F consisted of 46 MSME actors who did not receive fintech-based training on organizational support before the pandemic.

**Table 1.** SME Criteria.

| Business Size | Criteria | |
| --- | --- | --- |
| | Asset | Annual Revenue Sales |
| (1) | (2) | (3) |
| Micro business | Maximum of IDR 50 million | Maximum of IDR 300 million |
| Small business | >IDR 50 million–IDR 500 million | >IDR 300 million–IDR 2.5 billion |
| Medium Enterprises | >IDR 500 million–IDR 10 billion | >IDR 2.5 billion–50 billion |

Source: Bank Indonesia (2015).

The pre-test was carried out with 46 participants each in the control and experimental groups, with the following materials on Table 2.

**Table 2.** Training materials for fintech-based organizations before and during the COVID-19 pandemic.

| Time | Class Name | Content |
|------|------------|---------|
| (1) | (2) | (3) |
| Stage 1 | Regulation Fintech | 1. Regulation Fintech<br>2. Emerging Segments: RegTech, InsurTech, and LawTech<br>3. IoT and Cybersecurity |
| Stage 2 | Collaboration Fintech | 1. Collaboration Role in Financial Technology<br>2. Stakeholders Condition with Angel Investors, Fund Managers, Corporate Mentors<br>3. Policies Related to Regulation, Collaboration, and Financial Literacy |
| Stage 3 | Financial Literacy | 1. The Importance of Financial Literation in SME<br>2. The Concept of Fintech and Its Capabilities<br>3. Industry Convergence and Evolution of New Business Opportunities |

Source: Authors' calculations.

The experimental group was given training materials to support the organization. As described in Table 2, the main topics of this training are divided into 3 steps are regulation fintech, collaboration fintech, and financial literacy. The training was arranged by assigning 60 min to session 1, group discussions of predetermined topics, recommendations and practical implications. Others include (1) reduce costs of business operations, (2) improve customer service, (3) enhance distribution channels, (4) reap operational benefits, (5) provide effective support role to operations, and (6) increase ability to compete. This pattern was used in every training conducted, with a total of 9 fintech-based training meetings. The control group was not given preference because it ran a business according to the owners respective understanding before and during the pandemic. After the experiment, attention was directed towards owners in interpreting organizational support through observation and distributing questionnaires, which was later adopted as the post-test result.

*3.2. Data Treatment*

This is a quasi-experimental design, with the *t*-test method used to evaluate the effects of training and discussion of different models and the multivariate factor used for analysis. Organizational support provided through fintech-based training before and during the pandemic was carried out in detail with the following steps:

Step 1: Different tests were carried out on a sample group consisting of two varying conditions. Paired-samples *t*-test was conducted to determine whether there were significant differences with varying conditions in the same group (Chu and Choi 2000).

Step 2: Multivariate factor analysis was carried out to determine priorities in organizational support in various conditions. This technique is used to identify a small number of factors or latent constructs from a large number of observed variables (Hair et al. 2018; Worthington and Whittaker 2006). According to Ryan and Tipu (2013) the multivariate normality of the data underlies the assumptions for the factor analysis curve. Normality testing is performed by checking probabilities (P-P) and plot quantiles (Q–Q) for data in the current study for all variables following a linear formation that fits perfectly (Park 2008; Ryan and Tipu 2013). This process is carried out with the aim of obtaining certain groupings which theoretically have separate impact sizes with similarities (Björklund 2011; Hair et al. 2018).

The normalized data were then processed by PCA, which assessed the impact of changes in the value of the selected variables on the final result. PCA is a standard technique for simplifying datasets by extracting data for hidden features and relationships, while removing those with redundant information (Le et al. 2019). PCA assigns weights to each input variable included in the index construction with the first principal component determined as the new value which is considered the best representative (Le et al. 2019; Radovanović et al. 2018). Bartlett's test of sphericity and the Kaiser–Meyer–Olkin test (KMO) were performed at the start of the PCA to check the suitability of the factor analysis

data. Bartlett's sphericity test was used to evaluate whether the correlation matrix used in PCA is an identity type. It must be significant with a *p*-value less than 0.05 for a fit factor analysis (Le et al. 2019; Tabachnick et al. 2007). The first factor, or component, describes the greatest percentage of the total variation while the second explains the bulk of the remaining variants (Le et al. 2019; Radovanović et al. 2018) This extraction process continues until the number of identified component is equal to the original variable. After that, the components that describe a portion of the variance above a certain threshold is extracted and expressed (or eigenvalues) (Le et al. 2019; Radovanović et al. 2018).

## 4. Results

Small and medium enterprises (SMEs) are the driving force of developing countries' economies (Kartiwi and MacGregor 2011; Mittal et al. 2018). According to some preliminary studies, organizational efforts focus on a "digitization" strategy as the key to achieving Industry 4.0 capabilities. SMEs that successfully manage this digitalization transition will be successful in the future. Therefore, fintech-based businesses are increasingly developing in Indonesia with the integration of digitalization comprising vertical and horizontal value chain (Ahmedova 2015; Mittal et al. 2018). Small and medium enterprises (SMEs) have tremendous potential for flexible adaptation of the country's economic conditions and respond to changing markets (Ahmedova 2015). SMEs are required to survive by maintaining their high flexibility character because consumer needs and demands changed during the pandemic. Consumer behavior at this time strengthens the position of digitalization, including the fintech sector.

*Statistical Analysis*

Before performing the difference test and data analysis, a normality test was carried out by evaluating the plot quantiles (Q–Q). A visual examination of the Q–Q plot shows that the values are along the diagonal without substantial deviations (Hair et al. 2018). The results show that the data is normally distributed.

A different test was carried out on six predetermined sample groups to obtain five groups in pairs. The results of the paired-samples *t*-test in Table 3 were tested with the following criteria:

1.  Comparing t tables and t value

    - If t value < t table, then $H_0$ is accepted.
    - If t value > t table, then $H_0$ is rejected or $H_a$ is accepted.

    The value of t table for N (100) with df (99) is 1.98422 and t table for N (46) with df (45) is 2.01410.

2.  See the significance value

    - If the significance value > 0.05 then $H_0$ is accepted.
    - If the significance value < 0.05 then $H_0$ is rejected.

    The test results for pair 1 are t value $\leq$ t table ($-10.495 < -1.98422$) and significance ($0.000 < 0.05$). Therefore, $H_0$ is rejected, meaning that there is a difference in organizational support in fintech-based businesses before and during the pandemic. This finding supports hypothesis 1, stating that the test results for pair 2 have a t value $\leq$ t table of $24.024 > 1.98422$ and significance of $0.000 < 0.05$, hence $H_0$ is rejected. This means that there is a difference in organizational support in fintech businesses during the pandemic. In hypothesis 2, the test results for pair 3 have a t value $\leq$ t table of $8.480 > 1.98422$ and significance of $0.000 < 0.05$, hence $H_0$ is rejected. This implies that there is a difference in organizational support in fintech efforts in conditions before the pandemic. In Hypothesis 3, the test results for pair 4 have a t value $\leq$ t table of $15.820 > 1.98422$ and significance of $0.000 < 0.05$, hence $H_0$ is rejected. This indicates that there is a difference in organizational support in fintech efforts in conditions before and during the pandemic with training. This finding supports hypothesis 4, stating that test results for pair 5 have a t value $\leq$ t table of $4.446 > 1.98422$ and significance of $0.000 < 0.05$, hence $H_0$ is rejected. It signifies that there is a difference in

organizational support in fintech businesses in conditions before and during the COVID-19 pandemic without training. This finding supports Hypothesis 5.

**Table 3.** Results of paired-samples *t*-test.

| | Group | N | Mean | SD | t | df | Sig (2 Tailed) |
|---|---|---|---|---|---|---|---|
| | (1) | (2) | (3) | (4) | (5) | (6) | (7) |
| Pair 1 | Before the COVID-19 pandemic | 100 | 3.4417 | 0.44215 | −10.495 | 99 | 0.000 * |
| | During the COVID-19 pandemic | 100 | 3.9367 | 0.68730 | | | |
| Pair 2 | During the COVID-19 pandemic with fintech-based training | 46 | 4.5616 | 0.32080 | 24.024 | 45 | 0.000 * |
| | During the COVID-19 pandemic without fintech-based training | 46 | 3.3188 | 0.28290 | | | |
| Pair 3 | Before the COVID-19 pandemic with fintech-based training | 46 | 3.7319 | 0.40289 | 8.840 | 45 | 0.000 * |
| | Before the COVID-19 pandemic without fintech-based training | 46 | 3.1413 | 0.20777 | | | |
| Pair 4 | During the COVID-19 pandemic with fintech-based training | 46 | 4.5616 | 0.32080 | 15.820 | 45 | 0.000 * |
| | Before the COVID-19 pandemic with fintech-based training | 46 | 3.7319 | 0.40289 | | | |
| Pair 5 | During the COVID-19 pandemic without fintech-based training | 46 | 3.3188 | 0.28290 | 4.446 | 45 | 0.000 * |
| | Before the COVID-19 pandemic without fintech-based training | 46 | 3.1413 | 0.20777 | | | |

* $p < 0.05$. Source: Authors' calculations.

The factor analysis feasibility test met the requirements for the Kaiser–Meyer–Olkin Measure of Sampling Adequacy (KMO) value > 0.5 and Bartlett's test significance value < 0.05 (Hair et al. 2018). Table 4 shows that the overall value of KMO organizational support before and during the pandemic with and without training is greater than 0.05, hence this factor analysis is feasible to be carried out to the next stage.

**Table 4.** Results of Bartlett's test of sphericity and the Kaiser–Meyer–Olkin Measure of Sampling Adequacy.

| | Bartlett Test of Sphericity | | | Kaiser–Meyer–Olkin Measure of Sampling Adequacy |
|---|---|---|---|---|
| | Chi-Square | df | *p*-Value | |
| Organizational support for fintech businesses before the pandemic | 355.230 | 15 | 0.000 * | 0.832 |
| Organizational support for fintech efforts during the pandemic | 558.904 | 15 | 0.000 * | 0.896 |
| Organizational support for fintech businesses during the pandemic with fintech-based training | 83.614 | 15 | 0.000 * | 0.695 |
| Organizational support for fintech businesses during the pandemic without fintech-based training | 60.109 | 15 | 0.000 * | 0.566 |

* $p < 0.05$. Source: Authors' calculations.

The next stage is to determine the number of factors on Table 5 through eigenvalue. Organizational support in the fintech business before the pandemic resulted in 1 factor with an eigenvalue of 3.960, and a cumulative variance of 65.996%, meaning that it contributed to the commonality of 65.996%. Organizational support for the fintech business during the pandemic produced 1 factor with an eigenvalue of 4.705, and a cumulative variance of 78.409%, meaning that it contributed 78.409% to the community. Organizational support for

the fintech business during the pandemic with fintech-based training resulted in 2 factors with an eigenvalue of 2.682 and 1.431 for factors 1 and 2. This has a cumulative variance of 68.547%, which means that its contribution to community is 68.547%. Organizational support during the pandemic without fintech-based training resulted in 2 factors with an eigenvalue of 2.411 and 1.177 for factors 1 and 2. It also has a cumulative variance of 59.801%, meaning that it contributes to community 59.801%. Finally, the factor charge is determined by using Varimax rotation.

**Table 5.** Results of exploratory factor analysis.

| Scale and Item | Organizational Support for Fintech Businesses before the COVID-19 Pandemic | | Organizational Support for Fintech Efforts during the COVID-19 Pandemic | | Organizational Support for Fintech Businesses during the COVID-19 Pandemic with Fintech-Based Training | | Organizational Support for Fintech Businesses during the COVID-19 Pandemic without Fintech-Based Training | |
|---|---|---|---|---|---|---|---|---|
| | **Factor** | | **Factor** | | **Factor** | | **Factor** | |
| | **1** | **2** | **1** | **2** | **1** | **2** | **1** | **2** |
| OS1:Reduce costs of business operations | 0.663 | - | 0.799 | - | 0.019 | **0.497** | 0.025 | **0.951** |
| OS2:Improve customer service | 0.773 | - | 0.863 | - | **0.713** | 0.124 | **0.504** | 0.164 |
| OS3:Improve distribution channels | 0.763 | - | 0.004 | - | **0.905** | 0.135 | **0.719** | 0.006 |
| OS4:Reap operational benefits | 0.803 | - | 0.918 | - | **0.775** | 0.156 | **0.825** | −0.076 |
| OS5:Provide effective support role to operations | 0.790 | - | 0.824 | - | 0.203 | **0.501** | **0.289** | 0.230 |
| OS6:Increase ability to compete | 0.821 | - | 0.876 | - | 0.144 | **0.844** | **0.542** | 0.199 |
| Eigenvalue | 3.960 | - | 4.705 | - | 2.682 | 1.431 | 2.411 | 1.177 |
| Percentage of variance explained | 65.996 | - | 78.409 | - | 44.704 | 23.844 | 40.187 | 19.614 |
| Cumulative percentage of variance explained | 65.996 | - | 78.409 | - | 44.704 | 68.547 | 40.187 | 59.801 |

Source: Authors' calculations.

During the pandemic fintech-based training, factors closely correlated with factor 1 are improved customer service, enhanced distribution channels, and reap operational benefits. Others include market capitalizing reconfiguration (MCR), the factor that reduces costs of business operations, provides effective support, increases the ability to compete, and operational adjustment reconfiguration (OAR). However, those closely correlated with factor 1 without fintech-based training are improved customer service, enhanced distribution channels, reaped operational benefits, effective support role to operations, increased ability to compete, and strategic performance reconfiguration (SPR). The correlating factors are a decrease in costs of business operations, and suitable financial performance reconfiguration (FPR).

## 5. Discussion

The enormous economic and social impact of the pandemic has become a large-scale event with consequences, especially for MSMEs globally (Goodell 2020; International Trade Centre 2020). This has disrupted organizations, and therefore solutions are needed to maintain success amid the pandemic. Such a tool is the use of organizational support with significant differences before and after the pandemic due to fintech efforts (Chang and Chen

2015; Grandon and Pearson 2004; Ibrahim et al. 2016). Therefore, MSME owners need to focus on seeking changes in organizational support design in fintech businesses to be able to face this uncertain condition. This can also be adjusted to the behavior of consumers who use more electronic payment instruments. Therefore, organizations must be agile in facing the challenges that exist and capable of providing support when needed. These include reducing business operating costs, improving customer service, enhancing distribution channels, reaping operational benefits, providing an effective support role for operations, and improving the ability to compete. The concrete form is to provide training to enable employees to support the organization. Financial Technology must be mastered to achieve this process. This is in line with the TPB theory, stating that people display certain behavior when they think others have the same opinion (Ajzen 1991). Therefore, organizational support can lead to affective commitment through the development to be achieved, especially in fintech, which is a necessity. The economic impact for FinTech players in Indonesia is still dominant in doing business with the payment segment (43%), credit (17%), crowdfunding, and others. Organizational support generates a norm of reciprocity from employees to care and help achieve its goals. The consequence is an increase in positive attitudes toward work, such as affective commitment, job involvement, as well as increasing favorable behavior. Contributions given by employees to the organization will be reciprocal in the form of acceptance and recognition (Caesens and Stinglhamber 2014; Ibrahim et al. 2016; Jeung et al. 2017). Organizational support can be assumed to be the fulfillment of the socio-emotional needs of employees at work, such as the need for self-esteem, care, emotional support, and recognition. Hence, the consequences of fulfilling these needs increase the subjective well-being of employees, such as job satisfaction and health.

Fintech regulatory topics are needed to increase business scale in the current era, and therefore the study results can generally be adopted by other countries, especially due to the similarity. Fintech-based MSMEs are required to be able to run their organizations well amidst the many requests for financial technology during the pandemic (Sahay et al. 2020; Tut 2020). However, it is undeniable that MSMEs have constraints related to some financial knowledge (Sahay et al. 2020) and organizational support (Zhou 2016). MSME owners need to provide good fintech-based training because the statistical results show good differences between trained and untrained employees. In accordance with the factor analysis results, training needs to be designed with due regard to priority factors, namely market capitalizing reconfiguration (MCR) and operational adjustment reconfiguration (OAR). This is because digital-based training deepens financial literacy, regulation, and collaboration in the fintech ecosystem. According to preliminary studies, it is imperative for MSMEs to gather provisions regarding digital financial inclusion in order to expand access to finance for low-income households and small businesses (Sahay et al. 2020). This is in line with the research by (Guzzo et al. 1985) and Schraeder (2009) stating that training programs are the most powerful activity among many organizations. Training works to improve employee performance and achieve a competitive advantage for the organization. The goal is for employees to master the skills and behaviors emphasized in the program and their application in daily activities.

The ease with which MSMEs adopt fintech will make it easy for them to weather through difficult times associated with the pandemic easily. This is because digital financial services are faster, cheaper, and more efficient, hence they can increasingly reach low-income households and small and medium enterprises (SMEs). During the pandemic health crisis, digital financial services make contactless and cashless transactions possible. Therefore, it is believed that with digital financial inclusion, MSMEs will help facilitate the implementation of efficient and fast government support measures, including for people and companies affected by the pandemic (Sahay et al. 2020; Tut 2020). From an economic perspective, fintech is very useful because it accelerates monetary policy transmission, increases the speed of money circulation, and economic growth. Finally, with organizational support, MSMEs were able to enhance economic defense capabilities back during the pandemic.

## 6. Conclusions

In conclusion, there are differences in using fintech by MSMEs to increase their defense capabilities during the COVID-19 pandemic. The results also showed differences in organizational support when fintech-based training was given top priority factors, namely market capitalizing reconfiguration (MCR) and operational adjustment reconfiguration (OAR) with deepened financial literacy, fintech regulation, and collaboration. The importance of this training requires related parties to equip MSMEs with regulation, collaboration, and literacy training.

Practical implications of this research are the need for MSMEs to have organizational support through the following (1) decrease in costs of business operations, (2) rise in customer service, (3) increase in distribution channels, (4) reap operational benefits, (5) provision of an effective support role to operations and (6) increase in ability to compete. Training is important to ensure fintech regulation, collaboration, and literacy. Theoretical contributions are related to the forming factors of fintech-based organizational support, which can be studied further to be tested empirically, market capitalizing reconfiguration (MCR) and operational adjustment reconfiguration (OAR). Another theoretical contribution is that the situation and conditions of the pandemic can be a factor distinguishing the behavior of organizations from participating in fintech-based training.

The limitation of this research is the possibility of bias in the evaluation of monitoring and implementation of organizational policies in each fintech venture. This is because it is conducted traditionally with the perception of organizational success from each assessment team. Further research is necessary to improve the characteristics of respondents both in terms of business scale and research location. Recommendations also need to be discussed qualitatively in depth. The limitations of this research will actively support the results of subsequent research for it to be implemented in helping the development of MSMEs.

Theoretically, the findings of this study can be used as a reference for further studies on fintech-based SMEs in their defense capabilities amid the pandemic by optimizing organizational support. Practically, this research has implications for the community, especially fintech-based MSMEs. The recommended practical solution is to design training that requires regulation, collaboration, and literacy to improve economic stability amid the COVID-19 pandemic.

**Author Contributions:** All authors contributed to the study conception and design. Conceptualization, C.H.; methodology, C.H. and F.D.P.; software, F.D.P.; validation, C.H. and F.D.P.; formal analysis, C.H. and F.D.P.; investigation, C.H.; resources, C.H.; data curation, F.D.P.; writing—original draft preparation, C.H. and F.D.P.; writing—review and editing, C.H. and F.D.P.; visualization, F.D.P.; supervision, C.H.; project administration, C.H.; funding acquisition, C.H. All authors have read and agreed to the published version of the manuscript.

**Funding:** This research was funded by Universitas Ciputra Surabaya.

**Conflicts of Interest:** The authors declare no conflict of interest.

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
