# Peer review of "Impact of COVID-19 on Organizational Support in Financial Technology"

_economies, doi:10.3390/economies10080183_

Round 1
Reviewer 1 Report
Please find commets in the attachment.

Author Response
Dear. Reviewer 1
We attach a revision that has been done well. We are grateful for the revision of the notes provided. Please see in the attachment. We hope that this revision of our article will be well received. For the attention of the reviewers, we thank you very much.
Regards,
Dr Christian Herdinata

Reviewer 2 Report
Dear authors,
It is a pleasure to review this study. This paper is very well organized and supported. However, some changes are needed:
- In the abstract add the originality of the paper;
- Discussion of results is weak. All results must be discussed and substantiated in a deeper way based on the literature review carried out.
- Add and deepen in the discussion of results the practical and theoretical implications of this study. There are only minor implications in the conclusion, but they are not enough.
Author Response
Dear. Reviewer 2
We attach a revision that has been done well. We are grateful for the revision of the notes provided. Please see in the attachment. We hope that this revision of our article will be well received. For the attention of the reviewers, we thank you very much.
Regards,
Dr. Christian Herdinata

Round 2
Reviewer 2 Report
The authors carried out in-depth revisions with the suggestions that had been made. These changes significantly improved the quality of the paper.